

# Using deep learning-based artificial intelligence electronic images in improving middle school teachers' literacy

Yixi Zhai, Liqing Chu, Yanlan Liu, Dandan Wang and Yufei Wu

School of Foreign Studies, Tangshan Normal University, Tangshan City, China

Corresponding author
Yixi Zhai, zhaiyixi@tstc.edu.cn

## ABSTRACT

With the rapid development of societal information, electronic educational resources have become an indispensable component of modern education. In response to the increasingly formidable challenges faced by secondary school teachers, this study endeavors to analyze and explore the application of artificial intelligence (AI) methods to enhance their cognitive literacy. Initially, this discourse delves into the application of AI-generated electronic images in the training and instruction of middle school educators, subjecting it to thorough analysis. Emphasis is placed on elucidating the pivotal role played by AI electronic images in elevating the proficiency of middle school teachers. Subsequently, an integrated intelligent device serves as the foundation for establishing a model that applies intelligent classification and algorithms based on the Structure of the Observed Learning Outcome (SOLO). This model is designed to assess the cognitive literacy and teaching efficacy of middle school educators, and its performance is juxtaposed with classification algorithms such as support vector machine (SVM) and decision trees. The findings reveal that, following 600 iterations of the model, the SVM algorithm achieves a 77% accuracy rate in recognizing teacher literacy, whereas the SOLO algorithm attains 80%. Concurrently, the spatial complexities of the SVM-based and SOLO-based intelligent literacy improvement models are determined to be 45 and 22, respectively. Notably, it is discerned that, with escalating iterations, the SOLO algorithm exhibits higher accuracy and reduced spatial complexity in evaluating teachers' pedagogical literacy. Consequently, the utilization of AI methodologies proves highly efficacious in advancing electronic imaging technology and enhancing the efficacy of image recognition in educational instruction.

## INTRODUCTION

Amidst the rapid expansion of societal information, electronic educational resources have emerged as an integral cornerstone of contemporary educational paradigms. The traditional pedagogical framework continues to face challenges in accommodating the diverse learning needs of modern students and aligning with the dynamic methods of

information acquisition (*Vadim, 2018*). The widening gap presents specific challenges for secondary school teachers, who must engage in instructional activities within an educational environment continuously influenced by the transformation of electronic resources. As a pivotal component of the educational continuum, the professional competence and instructional efficacy of middle school educators exert a direct influence on students' learning outcomes and academic achievements. Cognitive literacy, encompassing a teacher's proficiency in leveraging advanced technologies to enhance instructional capabilities and adapt to diverse learning needs, emerges as a comprehensive competency. This proficiency spans various aspects of intelligent technology applications, including but not limited to data analysis, personalized instructional design, and real-time feedback. Consequently, enhancing the cognitive proficiency of middle school teachers stands as an imminent challenge within the contemporary educational landscape (*Weng et al., 2020*).

This imperative is underscored by shortcomings in the conventional approaches to teacher training and evaluation (*Gamazo & Martínez-Abad, 2020*). In contemporary society, middle school teachers play a crucial role in education. However, traditional teacher training and evaluation systems often prove inadequate in addressing the practical challenges middle school educators face. This issue underscores the difficulties within the educational system in meeting individual differences, reflecting the complexities of the teaching environment, and providing timely and effective feedback. For instance, traditional training methods frequently fail to fully account for middle school teachers' unique challenges across different subjects, schools, and student populations. The lack of personalized training programs makes it challenging for teachers to respond to diverse instructional needs. Additionally, existing evaluation systems often overly rely on singular standards and assessment methods, overlooking the diversity and innovativeness of middle school teachers in their teaching practices. This singularity limits a comprehensive assessment of teachers' overall competence. Furthermore, traditional evaluation methods commonly suffer from delayed feedback, making it difficult for teachers to receive timely and effective improvement suggestions during the teaching process. This not only hampers professional development but also impacts students' learning experiences. Moreover, traditional training and evaluation systems often fail to leverage modern technological tools fully, hindering the integration of advanced technologies to enhance the accuracy and comprehensiveness of assessments. The lack of technological support restricts in-depth analysis of teaching practices and the application of innovative methods. These challenges underscore the urgency and necessity of reform within the education system.

To address these challenges, the utilization of artificial intelligence (AI) holds profound significance in enhancing the cognitive literacy of middle school educators (*Fernández-Batanero et al., 2022*). Initially, AI technology proved instrumental in delivering personalized teaching resources and feedback, adeptly catering to the diverse demands of students (*Sancar, Atal & Deryakulu, 2021*). Through the incorporation of AI-generated electronic images, educators can access a wealth of instructional materials, including high-quality teaching videos, interactive textbooks, and tailored learning tasks, thereby elevating the quality and efficacy of their pedagogy. Secondly, the AI methodology offers an objective and precise assessment of teachers' instructional processes and outcomes through

algorithmic models, facilitating the provision of personalized training and improvement recommendations (*Zhuang & Liu, 2022*). These evaluative approaches assist educators in identifying and resolving challenges, fostering their professional development and overall growth.

This study aims to analyze and explore the practical significance of employing AI methods, particularly AI electronic images and the SOLO algorithm, to enhance the cognitive literacy of middle school teachers. Through the utilization of AI electronic images, the experiment seeks to provide personalized and real-time insights into teachers' performance, addressing the limitations of current traditional assessment methods. Furthermore, the application of the SOLO algorithm ensures a more nuanced and comprehensive evaluation of teachers' educational effectiveness, meeting the demand for more adaptive and insightful assessment methods in a dynamic educational environment.

The innovation inherent in this study manifests in several dimensions. Firstly, it delves into the application of AI electronic images in the training and instruction of middle school teachers, furnishing them with diverse teaching resources and tools to foster their professional development. Secondly, the utilization of the Structure of the Observed Learning Outcome (SOLO) algorithm underpins an intelligent classification and algorithmic application model. This model provides objective and precise methodologies for the evaluation of middle school teachers' teaching, thereby enhancing teaching quality and effectiveness. Additionally, the study conducts a comparative analysis with other widely employed classification algorithms, such as decision trees and support vector machines (SVM), offering valuable insights and guidance for the application of AI in education.

This study addresses several issues inherent in current teacher training and evaluation methods. Firstly, leveraging AI electronic images and deep learning technology, educators gain access to diverse teaching resources, encompassing high-quality teaching videos, personalized textbooks, and student works, thereby better aligning with students' learning needs and improving instructional effectiveness. Secondly, the SOLO-based intelligent classification and algorithmic application model impartially and precisely assess teachers' instructional processes and outcomes. It furnishes personalized training and improvement recommendations, thereby facilitating their professional growth and development.

The structural framework of this study unfolds as follows: 'Introduction' introduces the contextual background of the SOLO classification algorithm and AI. Section 'Recent Related Work' provides a review of recent literature on AI and intelligent converged devices. Section 'Application and Effect Evaluation Model of Teachers' Intelligent Literacy based on the SOLO Classification Evaluation' evaluates middle school teachers' intelligent literacy and teaching effectiveness using the intelligent classification models of SVM and SOLO. Section 'Results' systematically assesses the impact of enhancing teachers' quality. Section 'Conclusions' concludes through induction and summary. This study holds practical application value for augmenting the training outcomes related to the intelligent literacy of middle school teachers.

## RECENT RELATED WORK

### Recent research status of AI and SOLO classification algorithm

The SOLO algorithm is an educational assessment tool employed to gauge students' comprehension and application of knowledge across various cognitive levels. By analyzing the structure of students' work, it categorizes it into distinct levels, thereby providing an assessment of the depth of learning. Numerous scholars have conducted research on its application in the field of education. *Kaul, Enslin & Gross (2020)* conducted an investigation into the historical development of AI in the field of medicine. Leveraging machine learning and pattern analysis, machines can enhance their capabilities through experiential learning from provided datasets (*Kaul, Enslin & Gross, 2020*). The findings indicated a pivotal role for AI in addressing gastrointestinal diseases. *Lv et al. (2020)* focused on the integration of AI into the Industrial Internet of Things (IIoT) system (*Lv et al., 2020*). Their research involved an analysis of the credibility of the Internet of Things system, culminating in the proposal of an automated online evaluation method for network information reliability. This study holds practical reference value for advancing the seamless integration of AI and IIoT. *Zhang & Lu (2021)* scrutinized AI research's current status and future prospects. Their comprehensive overview of AI encompassed industry information integration, usage context, drivers, technologies, and applications, revealing the rapid development of AI and its transformative impact on lifestyles. *Finlayson et al. (2021)* underscored AI's potential in enhancing the accuracy of medical services. *Huang et al. (2022)* delved into the scientific underpinnings of AI and its assistance in the development of new antibacterial drugs. Their findings indicated that automated AI could significantly improve the efficiency of drug development (*Huang et al., 2022*).

Moreover, in the realm of the SOLO classification algorithm and the enhancement of secondary school teachers' professional capabilities, *Ryan & Stieff (2019)* explored the assessment chart for secondary school chemistry learning achievements (*Ryan & Stieff, 2019*). Their analysis assessed the relative effectiveness of formative assessment methods involving drawing, oral communication, and symbolic modes. Chemical reactivity drawings derived from dynamic visualization were studied, revealing that evaluation projects incorporating drawings could facilitate the implementation of richer knowledge models. Additionally, compared to commonly used symbols in chemical science, students exhibited more conceptual features in their drawings and written explanations. *Moallem (2019)* investigated the influence of learning outcomes, knowledge acquisition, and higher-order thinking skills (*Moallem, 2019*). The study discerned how the characteristics of each method affected learning outcomes through case-based reasoning and inquiry. The SOLO-based classification algorithm was found to enhance the efficiency of knowledge acquisition. *Luo et al. (2020)* focused on a scientific argumentation model for students based on the structure of observed learning outcomes. They administered measurement tasks before and after scientific argumentation and teaching intervention (*Luo et al., 2020*). The data analysis results indicated that, following the teaching intervention, students in the experimental group exhibited significantly better performance in evidence and refutation compared to the control group. *Litualy & Serpara (2020)* conducted an analysis of static learning

skills, German language learning effects, and the outcomes of German students using SOLO, employing observation and interviews (*Litualy & Serpara, 2020*). The findings demonstrated that accurate and reliable calculation results could be derived through the computation and analysis of the obtained data. *Prachagool (2021)* delved into project-based learning and learning outcomes in young children. The analysis of 25 young children's learning outcomes, employing both qualitative and quantitative methods, indicated a high understanding ability in managing literature learning and projects (*Prachagool, 2021*). The study suggested that different project management environments could be provided based on varying learning perceptions and potentials (*Prachagool, 2021*). *Sanjaya, Suartama & Suastika (2022)* investigated the impact of portfolio evaluation on students' learning outcomes in civic education. Employing a multi-stage random sampling technique to select 120 student samples, they utilized Analysis of Variance (ANOVA) to analyze the data. The results indicated that students employing the conflict resolution learning model outperformed those using the traditional model (*Sanjaya, Suartama & Suastika, 2022*).

*Timmons et al. (2023)* indicated that AI technology holds the potential to enhance the assessment, diagnosis, and treatment of individuals with mental health issues, thereby expanding the coverage and impact of mental health care. However, if AI applications are constructed based on historical data reflecting underlying social biases and inequalities, they may fail to mitigate mental health disparities. *Celik (2023)* asserted that due to the emerging ethical issues associated with AI, educators must also possess the knowledge to evaluate decisions based on AI. Therefore, a scale has been developed to measure knowledge in using AI in teaching based on technological, pedagogical, and content knowledge frameworks. *Alhumaid et al. (2023)* aimed to understand the perceptions of UAE users regarding the use of AI applications for educational purposes. The study results indicate that the performance of diffusion theory variables surpasses those of "doing business" and "technology export" variables. AI is user-friendly and possesses useful features that can be shared across a variety of services provided.

The aforementioned research reveals that conventional centralized training and qualitative evaluation methods fall short of meeting the individual needs of teachers and fail to provide objective and accurate assessments, consequently resulting in limited efficacy in teacher training and evaluation outcomes. Traditional teacher training is often constrained by temporal and spatial limitations, unable to furnish diverse and personalized training resources (*Mazzye, Duffy & Lamb, 2023*). The evaluation process heavily relies on subjective judgments, introducing potential inaccuracies and subjectivity. Methodologically, numerous studies employ qualitative analysis methods to explore issues related to teacher training and evaluation, emphasizing the necessity for increased objectivity and accuracy. Typically, uniform training content and evaluation criteria are adopted, proving insufficient to address the specific needs and differentiated development of individual teachers. This study introduces a novel teacher training and evaluation methodology utilizing AI electronic images and deep learning technology. Leveraging potent data processing and pattern recognition capabilities inherent in AI technology, the approach provides personalized teaching support and yields objective and accurate assessment results. The proposed research methodology tailors teaching resources and

training support to meet the unique needs and characteristics of individual teachers. Through the integration of AI electronic images and algorithmic models, teachers gain access to a diverse array of teaching resources and personalized training recommendations, thereby fostering their professional development. Furthermore, the study incorporates an intelligent classification and algorithm application model based on SOLO, enabling an objective and accurate evaluation of teachers' instructional processes and outcomes. In contrast to traditional qualitative evaluation methods, this study introduces more scientifically rigorous and precise evaluation indicators, aiding teachers in identifying and addressing challenges. As such, this study contributes to bridging existing gaps in the field and presents notable advantages.

## Integrating intelligent equipment and secondary school teachers' intelligent literacy

In the exploration of intelligent fusion device applications in pedagogical practices, *Bengtsson (2020)* employed game-based learning methodologies to enhance the overall efficiency of the device. The study scrutinized and introduced the game, its components, and learning objectives (*Bengtsson, 2020*), culminating in a comparative analysis of game-based learning methods and recommendations for future improvements. The research holds practical reference value in advancing learning efficiency. *Dong & Li (2021)* delved into sports teaching simulation systems employing wearable virtual reality devices, emphasizing teaching mode innovation to foster students' self-discipline, innovation abilities, and capabilities for educational reform, all facilitated by intelligent equipment. *Li & Su (2021)* investigated the application of data mining tools for English writing and teaching, offering insights into the research status of data mining in college English writing abroad. The study proposed future research content and methods, suggesting the use of association rule algorithms in data mining to analyze factors influencing students' writing achievements, providing decision-making suggestions for teachers' instruction (*Li & Su, 2021*).

*Yang (2018)* explored the art curriculum of preschool vocational education within the context of innovation and entrepreneurship education, assessing the improvement and application effects on the intelligent literacy of middle school teachers. Enhancing the educational curriculum's structure, instructional content, and pedagogical approaches within the domain of art education may serve as a fundamental initiative for advancing the overall instructional quality of the specialized educational program (*Yang, 2018*). *Wang (2019)* established a network structure for the protection of intangible cultural heritage, discussing strategies for safeguarding such heritage in education. Eight professionals engaged in the field of batik were interviewed, revealing that the integration of cultural heritage education with the exigencies of contemporary daily life and societal circumstances is imperative in the present context (*Wang, 2019*). *Meyer & Aikenhead (2021)* conducted an analysis of pedagogical practices in Canadian mathematics education, leading to the formulation and adaptation of curriculum plans intended for use by educators. The compilation of professional teaching materials and the augmentation of resources dedicated to mathematics instruction stand out as strategies that can contribute to the enhancement of educators' intellectual capabilities (*Meyer & Aikenhead, 2021*). *Iannucci, Richards &*

*MacPhail (2021)* investigated the interconnections between individual achievement, psychological resilience, and the multifaceted role conflicts experienced by educators. Employing parallel confirmatory factor analysis, the researchers substantiated the psychometric integrity of the proposed factor framework (*Iannucci, Richards & MacPhail, 2021*). The hypothesized model demonstrated a favorable fit with the empirical data. Subsequently, structural equation modeling was utilized to scrutinize and authenticate the theoretical relationships posited by the conceptual model. The findings substantiated the mediation model, affirming its robust fit with the observed data. *Yang et al. (2022)* investigated the multi-level regulatory effect of middle school campuses, revealing a relationship between teacher burnout and school atmosphere after controlling for various factors. These studies provide practical insights into promoting the intellectual literacy of middle school teachers.

Recognizing the centrality of teaching quality in education, educators focus on improving school teaching quality through the use of teaching quality evaluation systems. This study contributes to the existing research on the integration of intelligent devices and the intellectual literacy of middle school teachers. Prior studies have explored the utilization of intelligent devices, such as smartphones and tablets, for personalized teacher training. These investigations emphasize the personalization and flexibility of training content, offering expertise and teaching resources through mobile learning platforms and applications. Other studies concentrate on the use of intelligent devices and auxiliary tools, such as instructional management systems, smart whiteboards, and online learning platforms, to enhance teachers' intellectual literacy by facilitating the organization and communication of instructional content, as well as providing instructional analysis and feedback. This study uniquely focuses on the evaluation of teacher literacy and presents distinct advantages in integrating intelligent devices with the intellectual literacy of middle school teachers. By employing intelligent devices as tools for teacher training and evaluation, educators gain access to personalized teaching resources and training support, irrespective of time and location, thereby enhancing their professional abilities and teaching effectiveness.

# APPLICATION AND EFFECT EVALUATION MODEL OF TEACHERS' INTELLIGENT LITERACY BASED ON THE SOLO CLASSIFICATION EVALUATION

## Integration of AI electronic images in the training and pedagogy of middle school educators

With the evolution of social informatization, AI electronic images, as a sophisticated technological tool, possess significant potential in the realm of middle school teacher training and pedagogy. The specific application of AI electronic images in the training and teaching of middle school educators encompasses three primary facets:

(1) Development of electronic and personalized teacher training content

AI electronic images are employed to create electronic and personalized teacher training content. This involves transforming training materials into electronic imaging and multimedia formats, enabling educators to access training resources seamlessly through

online learning platforms or mobile learning applications. This mode of training delivers personalized learning resources and instructional support tailored to individual teacher needs and learning progress. By utilizing images, videos, and animations, educators can enhance their understanding of teaching concepts, methods, and subject knowledge. Additionally, the integration of electronic images facilitates online discussions and exchanges among educators, promoting collaborative professional growth.

(2) Utilization as an auxiliary tool for electronic imaging in the teaching process

AI electronic images function as auxiliary tools in the teaching process, enriching and enlivening instructional content. Educators leverage electronic imaging and multimedia resources to present abstract concepts, experimental procedures, simulated scenarios, and more, aiding students in comprehending and retaining knowledge effectively. For instance, 3D models, geographical maps, and biological tissue structures can be showcased using electronic imaging, allowing students to observe and interact, thereby deepening their understanding of complex subjects. Electronic images are instrumental in teaching demonstrations, experimental simulations, virtual practices, and other instructional activities, fostering a more hands-on and interactive learning experience for students.

(3) Application of electronic imaging in teacher evaluation and feedback

AI electronic images play a crucial role in teacher evaluation and feedback, facilitating a comprehensive understanding of teaching effectiveness and areas for improvement. Through the observation and analysis of the teaching process and students' learning responses, electronic imaging and data are utilized to quantitatively evaluate teaching performance. These images may include video recordings of lectures, students' learning records, and the use of teaching materials. AI technology is leveraged to automatically analyze and process these electronic images. Techniques such as image recognition and behavior analysis assess teachers' instructional behaviors and students' engagement, while natural language processing analyzes teachers' language expressions and students' feedback. This evaluative feedback provides teachers with objective insights and guidance, assisting in the identification of strengths and weaknesses and enabling adjustments to teaching strategies and methods, ultimately enhancing teaching quality and student performance.

In summary, the integration of AI electronic images into middle school teacher training and pedagogy offers diverse applications. It digitizes and personalizes training content, provides multimedia learning resources and online communication platforms, and fosters professional growth among educators. Furthermore, it serves as an auxiliary tool during the teaching process, presenting abstract concepts, simulating experiments, and offering interactive experiences to enhance student interest and understanding. Additionally, it facilitates teacher evaluation and feedback, providing objective data and guidance for continuous improvement in teaching methods and overall instructional quality. This study explores the profound impact of AI electronic images on middle school teacher training and teaching, offering critical insights and references for advancing educators' intellectual literacy and teaching effectiveness.

## Application and evaluation model of teachers' intelligent literacy based on SOLO classification

Utilizing digital imagery as its foundation, this investigation posits a model for the application and evaluation of teachers' intelligent literacy predicated on the SOLO classification method. The objective is to quantitatively assess the cognitive proficiency of middle school educators and furnish tailored instructional feedback. The SOLO classification method employed herein constitutes a deep learning-based image classification algorithm adept at accurately discerning teachers' pedagogical behaviors and performances. This algorithm achieves precision through feature extraction and learning from electronic imaging data. The study seeks to develop an intelligent classification model and algorithmic application utilizing the SOLO classification method. This involves amalgamating electronic image data from teacher training and the instructional process to assess the intelligent literacy of middle school educators.

In a targeted approach, this framework initiates the acquisition of electronic image data during the instructional activities of educators, encompassing teaching videos, students' learning records, and the utilization of instructional materials. Subsequently, the SOLO classification method is employed to extract and comprehend the distinctive features within these electronic images, facilitating the establishment of a classification model for evaluating teachers' intelligent literacy. This model autonomously discerns various aspects of teachers' instructional aptitude, such as their expressive capabilities, adaptability in employing teaching strategies, and interactions with students. The schematic representation of this model is presented in Fig. 1.

In Fig. 1, the model is divided into three layers: the application layer, the business logic layer, and the physical layer. At the application layer, the focus is on practical applications and operations. The Data Import module is responsible for importing teachers' instructional data into the system, including electronic images, audio recordings, and more. This ensures that the assessment model has sufficient information for accurate analysis. The Teaching Mode component is dedicated to capturing teachers' behaviors and performances in different scenarios. Through a comprehensive analysis of teaching patterns, the model can better understand teachers' instructional strategies and methods. The Overall Discourse Analysis section involves an in-depth analysis of the overall instructional discourse. By combining technologies such as speech recognition, SOLO classification, and behavior coding, this study can gain a more comprehensive understanding of interactions and information transmission during the teaching process. The business logic layer consists of several components: active audio detection, speech segmentation, SOLO classification, behavior encoding, and data statistics. At the physical layer, this study's attention is directed towards the hardware and underlying architecture of the system. This includes the devices, sensors, and network structures that support the operation of the entire assessment model.

Utilizing records from classroom teaching sessions, students' learning processes are systematically classified into pre-structural, single-structural, multi-structural, associative structural, and extended structural levels through the implementation of the SOLO classification method. This method enables the observation and categorization of students' performances in learning tasks, facilitating an assessment of their learning levels and

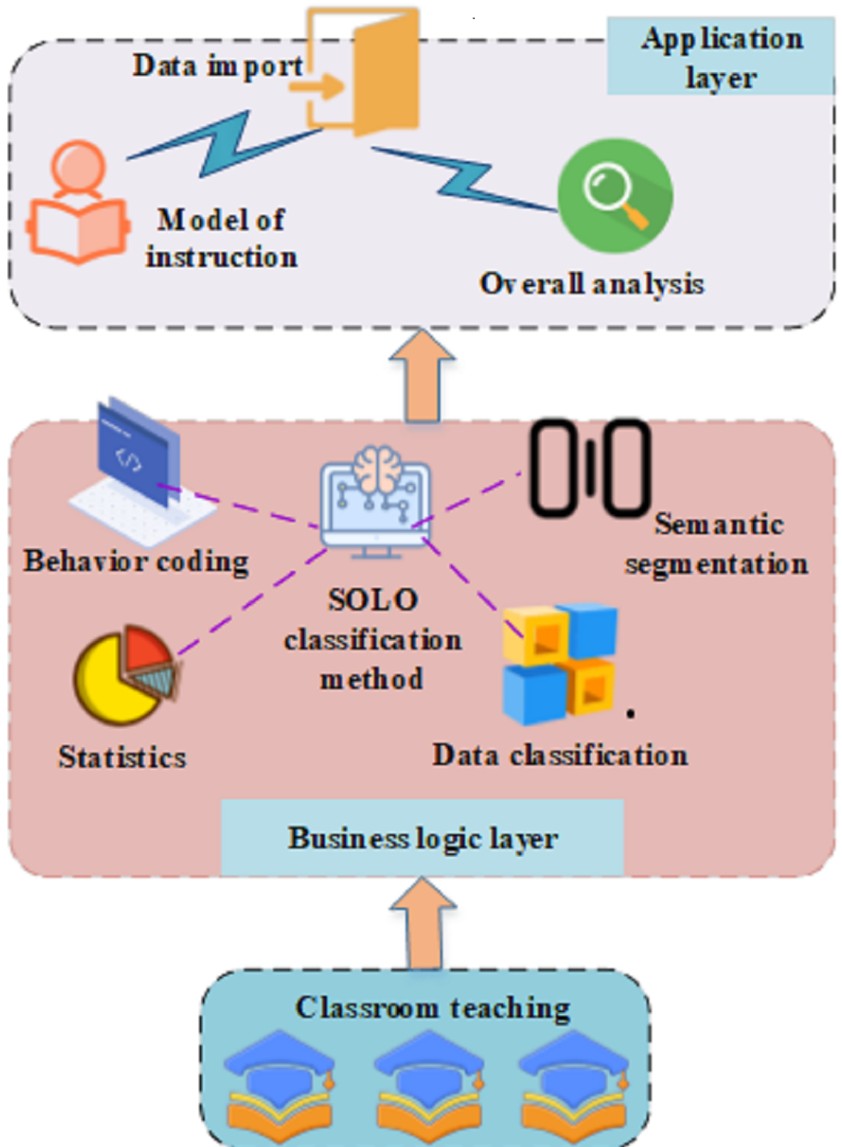

**Figure 1** Assessment model of teachers' intelligent literacy based on the SOLO classification.

competency development. To a certain extent, the effectiveness of students' learning serves as a reflection of teachers' instructional effectiveness. Within this study, the SOLO classification method is also employed to evaluate the intelligent literacy of middle school teachers. This approach aids in determining the teachers' positioning within the teaching process, offering personalized support and improvement suggestions.

The SOLO-based classification and improvement model of teachers' intelligent literacy stands as an efficient evaluation tool grounded in students' learning processes, summarizing the quality and depth of their learning experiences (*Qiu et al., 2022*). When assessing students' learning, teachers can discern their thought processes by examining their approaches to problem-solving and cognition. The SOLO classification method

categorizes cognitive levels into five tiers: pre-structural, unistructural, multistructural, relational, and extended abstract. Each level embodies distinct thought processes, reflecting a progression from simplicity to complexity, capturing the qualitative refinement of intelligent literacy. Progressing from structural to multi-structural levels, the enhancement of teachers' intelligent literacy is primarily manifested in elevating the structural foundation of knowledge. As knowledge extends across multiple levels to form an integrated structure, emphasis is placed on quantifying knowledge and exploring relationships between diverse knowledge elements. Advancing from relational design to abstract structure, this study investigates whether students can synthesize comprehensive disciplinary knowledge through the application and resolution of problems.

During the model application stage, electronic images of teachers are fed into the pre-trained SOLO classification model to derive assessments of teachers' intelligent literacy. The model outputs the intelligent literacy levels of teachers across various dimensions, including teaching proficiency, classroom management, student care, and more. Concurrently, the model generates personalized teaching feedback and recommendations, aiding teachers in comprehending their strengths and weaknesses and offering corresponding improvement measures.

In comparison to traditional teacher evaluation methods, the application and evaluation model of teachers' intelligent literacy based on SOLO classification in this study exhibits several advantages (*Waldia et al., 2023*). Firstly, the incorporation of AI and deep learning technologies facilitates the automatic extraction of key teacher characteristics from an extensive dataset of electronic images, mitigating the subjectivity and inaccuracy inherent in subjective evaluations and manual annotations. Secondly, the model construction, rooted in SOLO classification, enables refined assessments and personalized feedback regarding teachers' intelligent literacy, providing targeted instructional support and developmental guidance. Lastly, the model's application is characterized by real-time flexibility, allowing teachers to engage in self-evaluation and reflection at their convenience. Simultaneously, the model serves as an effective tool for schools and educational authorities to conduct comprehensive evaluations and monitor the literacy levels of the teaching force, enabling the implementation of targeted training and promotion programs.

In conclusion, the application and evaluation model of teachers' intelligent literacy, based on the SOLO classification method, holds significant importance in the realm of middle school teachers' training and teaching. In this study, the output of the SOLO algorithm is employed as the training target for the neural network. Specifically, the experiment applies the SOLO method to electronic image data depicting teacher behaviors and performances, guiding the neural network training based on the evaluative results generated by the SOLO algorithm. Throughout the training process, the single evaluation levels produced by the SOLO algorithm are utilized to enhance the neural network's ability to accurately capture the levels of teachers' intelligent literacy. By incorporating SOLO outputs as the training objective, the experiment enables the neural network to concentrate on crucial learning structures, thereby more effectively elevating teachers' intelligent literacy.

The SOLO method plays a pivotal role in training the neural network, directly influencing key steps such as weight updates and loss function calculations. Specifically, the output of

SOLO serves as a supervisory signal during training, guiding the neural network toward adjustments that align with the expected learning structures. Regarding weight updates, the neural network adjusts its parameters based on the evaluative results from SOLO, aiming to accurately reflect the levels of teachers' intelligence literacy. This contributes to preventing overfitting or underfitting of the model, enhancing the accuracy of the evaluation. In terms of loss function calculation, the output of the SOLO method is compared with the actual results, forming the foundation of the loss function. By minimizing the loss function, the study enables the neural network to learn and comprehend the learning structures of teachers' intelligent literacy, resulting in more precise evaluation and improvement. In this manner, the SOLO method actively participates in the training process of the neural network, providing a robust theoretical foundation and practical support for the effective assessment and enhancement of teachers' intelligent literacy. It harnesses the strengths of AI electronic images for the quantitative assessment of teachers' intelligent literacy, delivering personalized teaching feedback and recommendations. The utilization of this model holds the potential to stimulate teachers' professional development, elevate teaching quality, and contribute to the ongoing progress of middle school education.

## Classification algorithm and teaching quality evaluation based on SVM

The SVM-based classification algorithm was initially employed directly in the domain of multi-classification and has found practical applications in real-life scenarios such as text classification and face recognition (*Ghazal & Abdullah, 2020*). Currently, two primary methods exist for addressing multi-classification challenges using SVM technology. The first method involves the direct alteration of the objective function, transforming the multi-classification problem into a singular optimization problem, which is then resolved. The second method is indirect, where multiple SVM binary classifiers are initially established. Subsequently, combinatorial techniques are employed to assemble these classifiers into multiplexing systems supportive of vector devices. Essentially, SVM tackles multiple classification problems by amalgamating numerous binary classifiers. This amalgamation of binary subclasses into a dual-column structure not only resolves non-regional issues of vector machines but also enhances the program's classification efficiency and accuracy. Knowledge and historical data from distance education are subjected to analysis and deepening through data mining methods to generate knowledge and enhance the quality and service standards of distance education. Furthermore, this data aids teachers, students, and distance educators in elevating the quality assessment level. The evaluation of teacher professional literacy and competence development models is conducted based on the SVM. In addition to the proposed application and evaluation model of teachers' intelligent literacy utilizing the SOLO classification, this study introduces a classification algorithm and teaching quality evaluation method based on SVM (depicted in Fig. 2). SVM stands as a widely utilized machine learning algorithm in pattern recognition and classification problems. In this study, the SVM algorithm is incorporated to assess teachers' intelligent

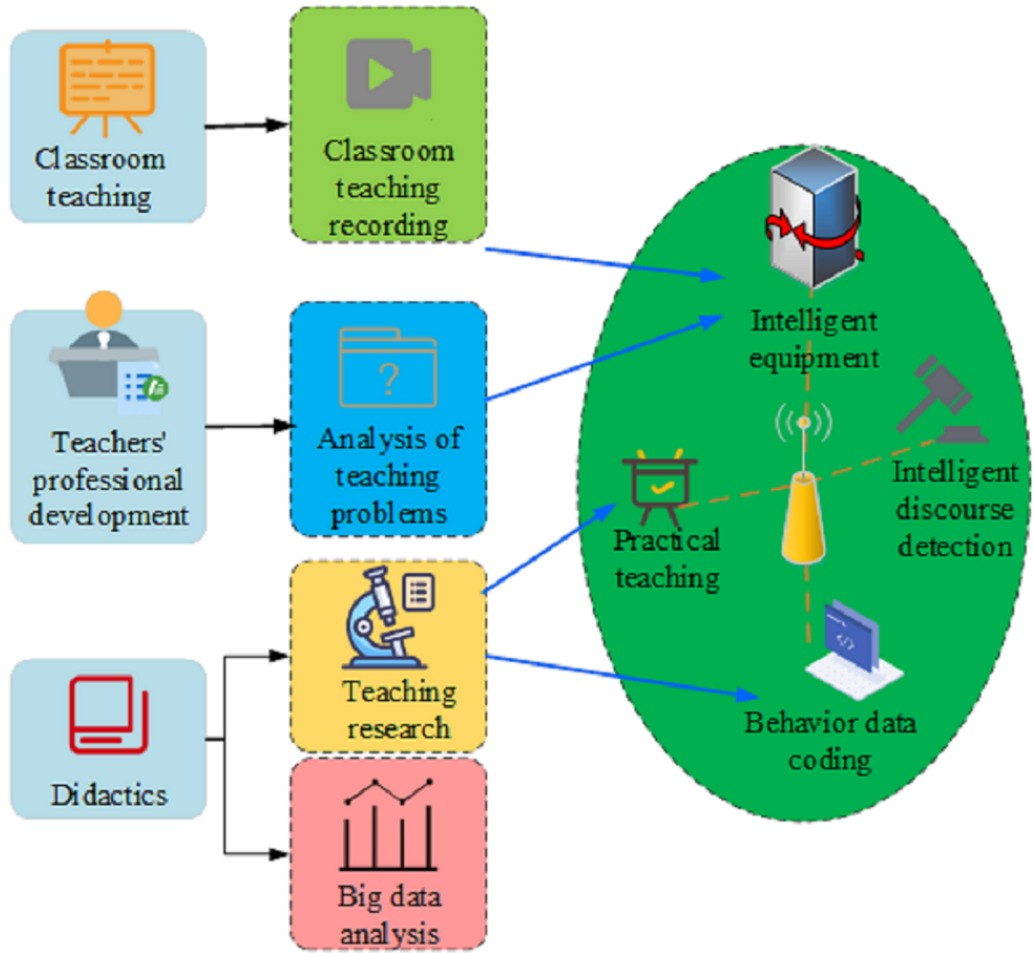

**Figure 2** **SVM-based teacher professional quality and ability improvement model.**

literacy, with the objective of exploring the effectiveness and advantages of different algorithms in teacher evaluation.

In Fig. 2, the SVM algorithm is employed to construct a model for the classification of teachers' intelligent literacy. Initially, data pertaining to teachers' intelligent literacy, encompassing teaching evaluation indicators and background information, is gathered. This data undergoes feature extraction and preprocessing to render it suitable for classification. Subsequently, the SVM algorithm is utilized to train and model the processed data, resulting in a model capable of classifying teachers' intelligent literacy. The teaching quality of educators in the classroom is characterized by their classroom teaching ability, professional competence, and knowledge of teaching theory. In this model, SVM records the teacher's classroom performance, which is then intelligently analyzed. The conversations are classified, and the behavioral data of teachers is coded and visualized. Through the analysis of big data in teaching, further research transforms into new teaching theories aimed at assisting teachers in enhancing their educational and teaching literacy.

## Teaching quality evaluation and research on teaching effect based on decision tree

To enhance teachers' understanding of students' learning content, specifically their knowledge base, and ideas, it becomes imperative to conduct teaching quality evaluations and explore teaching effects (*Lakeman et al., 2022*). Tasks such as classwork and exams can readily provide insights into a student's level of knowledge. However, the current state of students' thinking cannot be directly measured. In these scenarios, teachers need to analyze practical problems according to classification results, evaluate teaching quality through scientific decision-making, and assess students' responses to complex questions. It should be noted that most students employ a centralized and multi-point thinking method. The recursive method is applied during the generation of the decision tree, leading to the subdivision of the maximum information gain attribute. As a result, an equal number of sub-tables are created as attribute values, contributing to increased complexity in storage space when forming the decision tree. The structure of the teaching quality evaluation model is analyzed, and the structure is depicted in Fig. 3:

In Fig. 3, the model for evaluating teaching quality and conducting research on teaching effects, based on the decision tree, employs the decision tree algorithm as a method for assessing teaching quality and analyzing teaching effects. The model encompasses several crucial components:

(1) Data collection and feature selection: The model is required to collect teaching-related data, encompassing students' academic performance, engagement, homework completion, and other relevant metrics. These data serve as input features for the model. During the feature selection stage, meaningful features for teaching quality evaluation and teaching effect analysis are selected based on domain knowledge and actual requirements.

(2) Construction of the decision tree: The evaluation model is constructed using the decision tree algorithm, leveraging the collected data and selected features. The decision tree, structured as a classification and regression algorithm, segments the dataset into various subsets, allowing for judgments and predictions based on feature attributes.

(3) Feature weights and node division: Throughout the decision tree construction process, the weights of features are calculated, and nodes are divided into distinct branches based on these weights. Feature weights reflect their significance in evaluating teaching quality and analyzing teaching effects.

(4) Decision rules and prediction results: The decision tree model, formed by feature weights and node divisions, generates a set of decision rules. Adhering to these rules, the model predicts teaching quality and analyzes teaching effects based on input eigenvalues.

(5) Teaching quality evaluation and effect analysis: The constructed decision tree model is employed for evaluating teaching quality and analyzing teaching effects. By inputting corresponding eigenvalues, the model yields teaching quality grades or predictions of teaching effects, facilitating teachers in understanding the strengths and weaknesses of their teaching and identifying areas for improvement.

The research model for teaching quality evaluation and teaching effect based on a decision tree assesses teaching quality and analyzes teaching effects by constructing a decision tree, leveraging the distinctive characteristics of the decision tree algorithm.

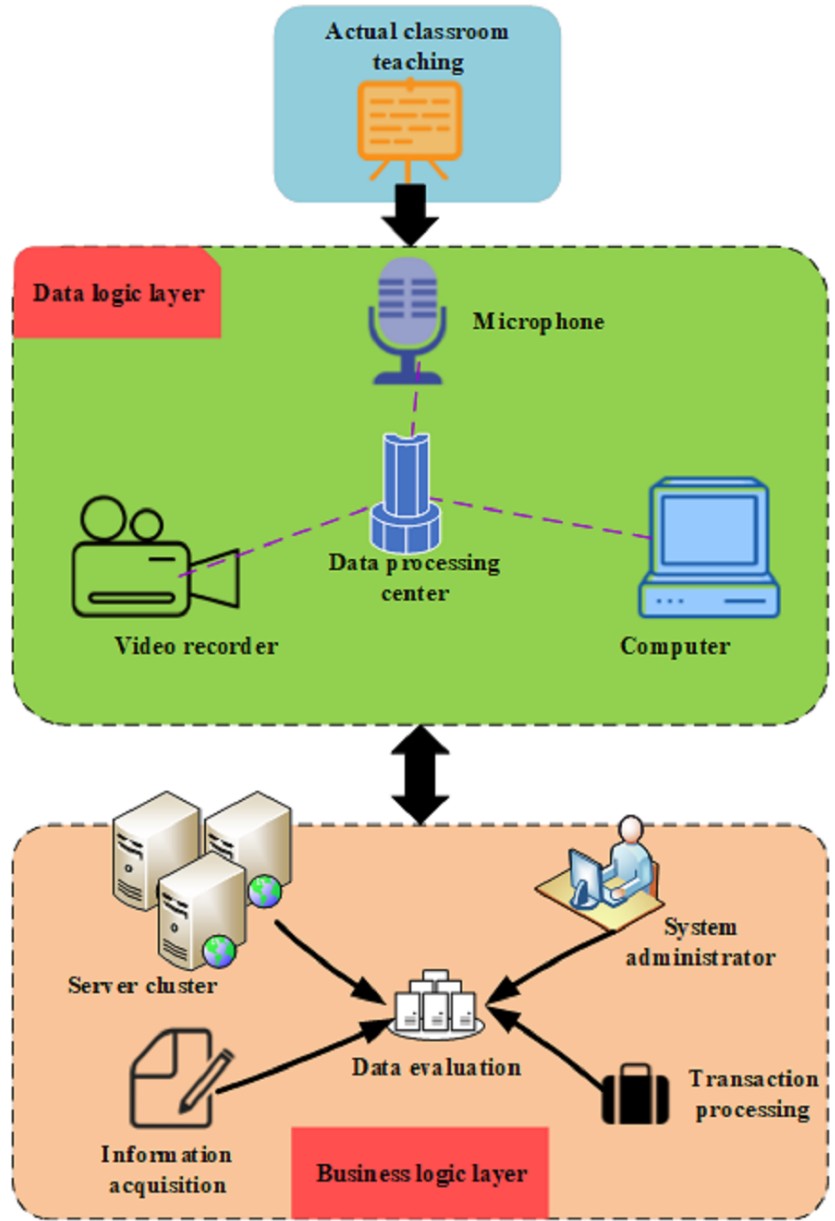

**Figure 3** Teaching quality evaluation model based on decision tree.

This model transforms intricate teaching data into transparent decision rules, aiding teachers in comprehending and applying evaluation results to enhance teaching quality and effectiveness.

## Application of AI electronic images

This study maximizes the use of AI electronic images to enhance the intelligent literacy of middle school teachers. This subsection will delve into the specific applications of AI electronic images in training content, teaching processes, and assessment/feedback.

(1) Application of AI electronic images in training content

The application of AI electronic images in training content extends beyond traditional training material provision to focus on personalized training experiences. By analyzing teachers' learning behaviors, individual differences, and specific teaching needs, the system can tailor training materials and courses for each teacher. This includes offering personalized learning paths, conducting in-depth analyses of teachers' challenges, and recommending relevant training resources, thereby catering to the diverse teaching backgrounds and needs of individual educators.

(2) Application of AI electronic images in the teaching process

During the teaching process, AI electronic images provide comprehensive support by capturing real-time teaching behaviors and student reactions. The system analyzes teachers' language expressions, teaching methods, and student interactions. The system can offer targeted suggestions through instant data feedback, including adjustments to teaching strategies and identification of student focal points, assisting teachers in addressing classroom challenges and improving teaching effectiveness.

(3) Application of AI electronic images in assessment and feedback

In terms of teaching assessment, the application of AI electronic images elevates teacher evaluation to a more precise and comprehensive level. The system, by analyzing factors such as teachers' teaching skills, student engagement, and classroom interactions, generates detailed assessment reports. These reports not only provide quantitative assessments of teaching effectiveness but also offer immediate feedback to teachers, assisting them in adjusting teaching strategies for more effective instruction.

Through these detailed application scenarios, this section elucidates the practical uses of AI electronic images in teacher training and teaching assessment, demonstrating their multi-dimensional and comprehensive supportive role.

## Experimental validation

Data preprocessing stands as a critical phase within the data mining process (*Gupta & Chandra, 2020*). By conducting meticulous and judicious data screening, the development of mining models can conserve substantial resources and enhance the accuracy of model evaluation outcomes. Prior to initiating the data mining process, considerations regarding the type and size of data become imperative. Consequently, the collected data pertinent to the intelligent literacy dataset of middle school teachers undergoes preprocessing, as delineated in Fig. 4. Typically, data mining constitutes only a segment of the overall database process. Initially, pertinent records are selected, followed by the input of multiple data components into learning activities within the database. Finally, the resultant output data is integrated back into the initial relational table for comprehensive results analysis. In order to assess the impact of the established models on improving teachers' intelligent literacy, this study adopts an experimental design based on the SOLO classification evaluation algorithm and intelligent teaching devices. The investigation explores the enhancement effects on middle school teachers' intelligent literacy and the quality of practical application. In designing the experiment to evaluate the influence of the established models on enhancing teachers' intelligent literacy, the study combines AI

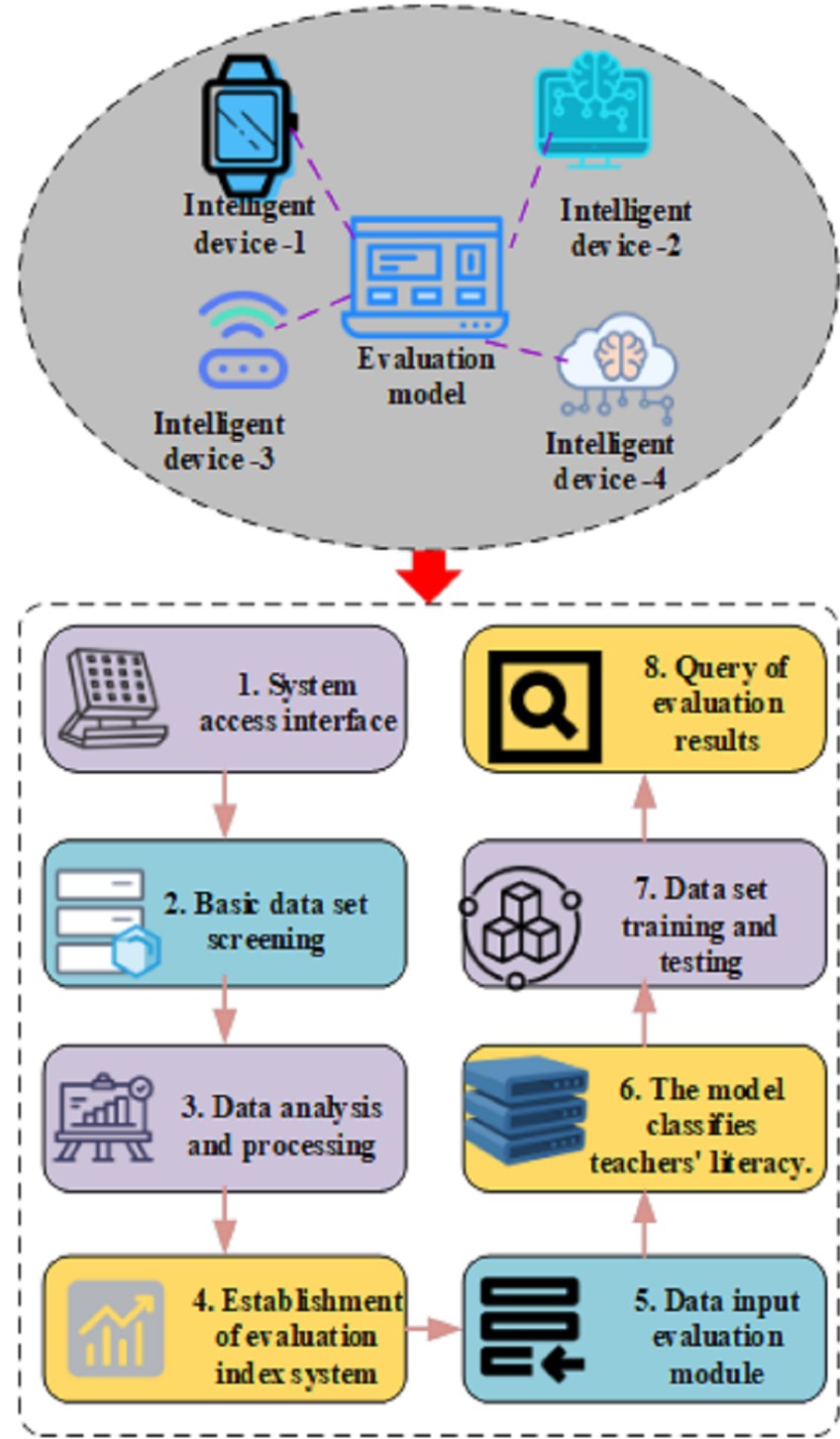

**Figure 4** Analysis of experimental steps of teaching quality evaluation model.

electronic images with the SOLO classification evaluation algorithm. This choice is driven by the intrinsic capabilities of AI electronic images to provide detailed insights into teachers'

performances, aligning with the assessment and improvement of middle school teachers' intelligent literacy. The fusion of AI electronic images and the SOLO classification method enables a more nuanced and comprehensive analysis of teachers' educational effectiveness, addressing limitations associated with traditional evaluation methods.

The experiment collected 1,000 electronic images of teachers across diverse teaching scenarios, capturing their instructional behaviors and performances. These images served as the training and testing sets for model construction and evaluation. AI electronic images refer to images processed and analyzed using AI algorithms. Leveraging these artificially enhanced images, the study aims to offer personalized and real-time insights into teachers' performances.

The dataset originated from teachers with various educational backgrounds and geographical locations, ensuring the model's broad applicability. The dataset's scale covers a sufficient number of samples to ensure thorough model training and testing. In terms of features, information encompassing teacher posture, facial expressions, student engagement, and other aspects of the teaching context was captured in the experiment. Rigorous quality control measures were implemented during data preprocessing, including noise removal and standardization of image brightness and contrast, ensuring the quality and consistency of the training data.

During the neural network training process, the experiment utilized the TensorFlow deep learning framework and set identical experimental parameters for the intelligent classification models based on SOLO and SVM. The iteration count was set to 600, the learning rate to 0.001, and the batch size to 32. In order to provide a detailed description of the neural network's training process, the study employed the convolutional neural network architecture, utilized the Adam optimization algorithm, and adjusted the model in each iteration to maximize performance. Such detailed explanations aid readers in fully understanding the methods and decision-making processes involved in model construction and training.

The pseudocode in Table 1 outlines the basic structure and operational flow of the SOLO algorithm:

Furthermore, 80% of the data was utilized as the training set for the SVM model, with the remaining 20% serving as the test set for performance evaluation to ensure the model's generalization capability across diverse datasets. The experiment systematically records the model's accuracy, training time, testing time, time complexity, and space complexity. Accuracy reflects the model's precision in identifying teachers' literacy, while time and space complexity provide insights into the efficiency and computational resource requirements for model implementation. These metrics collectively contribute to a comprehensive assessment of the model's performance. Through a comparative analysis of the SOLO-based and SVM-based intelligent classification models on these performance indicators, accuracy serves as a metric to evaluate their effectiveness and advantages in assessing teachers' intelligent literacy. The experiment opted for SVM and decision trees as comparative algorithms due to their prevalence in educational research and their advantages in handling classification tasks. Time complexity: It quantifies the temporal resources required for algorithm execution and signifies the growth rate of the algorithm's

**Table 1  Basic structure and operational flow of the SOLO algorithm.** The pseudocode in Table 1 outlines the basic structure and operational flow of the SOLO algorithm.

**# Pseudocode for SOLO Algorithm**

```
class SOLOAlgorithm:
    initialize_parameters()
        function train(data, labels, iterations):
            for epoch in range(iterations):
                for i in range(len(data)):
                    forward_pass(data[i])
                    compute_loss(labels[i])
                    backward_pass()
                    update_parameters()
    function predict(new_data):
        predictions = []
        for i in range(len(new_data)):
            output = forward_pass(new_data[i])
            predictions.append(output)
        return predictions
    function initialize_parameters():
        # Initialize network parameters, such as weights, biases, etc.
    function forward_pass(input_data):
        # Perform forward pass, compute output
    function compute_loss(true_labels):
        # Calculate the loss function by comparing output with true
        labels
    function backward_pass():
        # Backward pass, compute gradients
    function update_parameters():
        # Update network parameters, e.g., using gradient descent
```

running time concerning the problem size (*Li, Lin & Zhao, 2021*). In this context, time complexity is assessed by scrutinizing the execution times of operations, such as loops and conditional statements, within the algorithm. Spatial Complexity: It gauges the spatial resources consumed by an algorithm during its execution (*Wang et al., 2021*). This metric represents the growth rate of additional space necessary for the algorithm in relation to the problem size. The estimation of spatial complexity involves analyzing the data structures, variables, and storage employed in the algorithm.

The experimental environment settings are exhibited in Table 2:

Additionally, the experiment assessed the correctness of the course, employing a series of specific criteria to ensure a comprehensive, objective, and robust evaluation of teaching effectiveness. The detailed explanations and justifications for the evaluation criteria are as follows:

(1) Achievement of teaching objectives:

The experiment adopted the achievement of teaching objectives as a pivotal criterion because it directly reflects whether the course successfully conveyed the anticipated

**Table 2  The experimental environment settings.**

| Experimental environment settings | Settings |
|---|---|
| Hardware | GPU: NVIDIA GeForce GTX 1080 Ti (8GB video memory) |
| | CPU: Intel Core i7-8700K (6 kernels, 12 threads) |
| | Memory: 32GB DDR4 RAM |
| | Storage: 1TB SSD |
| Software | Deep learning framework: TensorFlow (version 2.0) |
| | Programming language: Python (version: 3.7) |
| | Optimization algorithm: Adam |
| | Comparison algorithms: SVM, decision tree |
| | Operating system: Windows 10 (64 bit) |

knowledge and skills. With meticulously designed teaching objectives, these goals aim to prompt students to attain clear academic outcomes during the course. This choice is grounded in research within the fields of education and curriculum design, emphasizing the importance of well-defined objectives in enhancing teaching quality.

(2) Student engagement:

Student active participation is another key criterion directly related to students' understanding of course content and their level of interest. The experiment quantified student engagement by observing and recording behaviors such as asking questions, answering queries, and participating in group activities. The selection of this criterion is based on the notion that highly engaged students are more likely to achieve good academic results and long-term learning outcomes.

(3) Quality of classroom interaction:

The experiment focused on the quality of classroom interaction because effective interaction can facilitate knowledge transfer and the establishment of a conducive learning atmosphere. This study assessed communication between teachers and students, interactions among students, and the quality of discussions related to course content. This choice is informed by research in educational psychology and interactive teaching, emphasizing the significance of positive classroom interaction in knowledge co-construction and subject understanding.

## RESULTS

### Performance analysis of recognition accuracy of teacher literacy improvement effect

This study conducts a comparative analysis between the SOLO-based model and the SVM-based algorithm, examining their respective performances and scrutinizing the accuracy of literacy recognition across different algorithmic models. The compiled data on recognition accuracy and the performance accuracy of distinct algorithms are presented in Figs. 5 and 6, respectively.

Figure 5 illustrates the progressive increase in the accuracy of teacher literacy recognition achieved by the SVM and SOLO algorithms as the model undergoes iterations. At 100 model

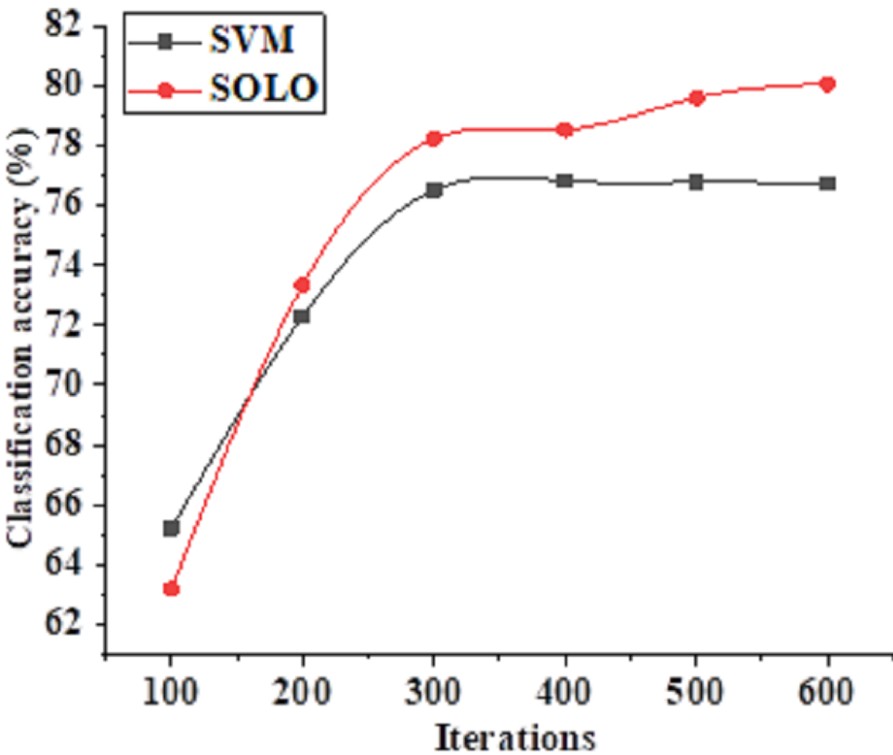

**Figure 5** Change curve of the accuracy of teacher literacy recognition based on different algorithm classification models.

iterations, the accuracy of teacher literacy recognition for the SVM and SOLO algorithms stands at 65% and 63%, respectively. Subsequently, as the number of model iterations reaches 200, the recognition accuracy of the SOLO algorithm surpasses that of the SVM algorithm. Upon reaching 600 model iterations, the accuracy of teacher literacy recognition for the SVM and SOLO algorithms reaches 77% and 80%. Consequently, the SOLO-based classification algorithm demonstrates superior accuracy in assessing the effectiveness of teachers' quality improvement, thereby exhibiting a more favorable application outcome.

In Fig. 6, the recognition accuracy of both SVM and SOLO models demonstrates an ascending trend with an increase in the number of model iterations. At the 100th iteration, the SVM model achieves a teacher literacy recognition accuracy of 69%, while the SOLO model attains a slightly higher accuracy of 70%. As the number of model iterations advances to 300, the accuracy of SVM literacy recognition improves to 76%, whereas the accuracy of the SOLO model further increases to 80%. Moreover, at 600 model iterations, the accuracy of teacher literacy recognition based on the proposed SOLO algorithm reaches 85%. Thus, the adjustment of the iteration count allows for fine-tuning the application effect of the model to attain an optimal value.

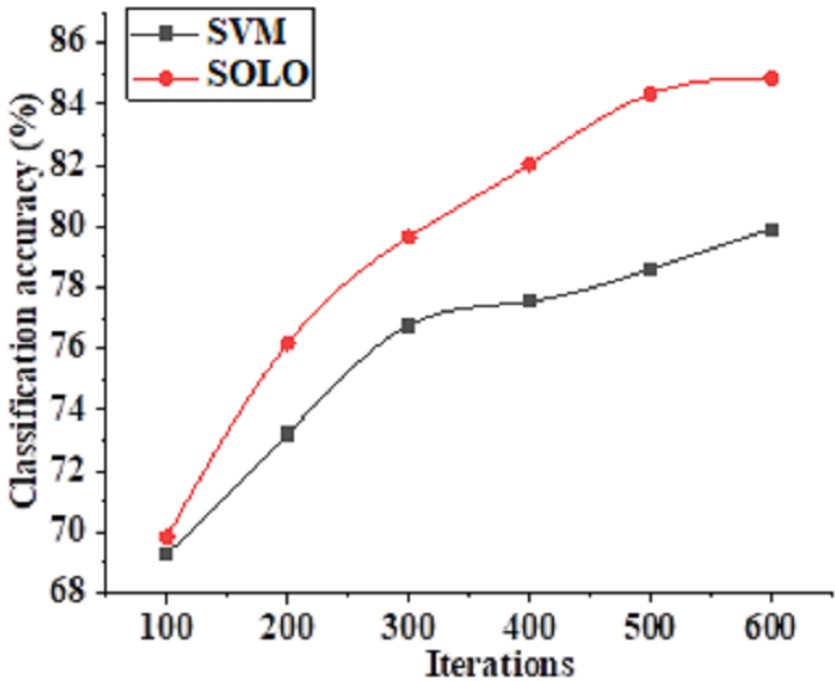

**Figure 6** Evolutionary trajectory of accuracy in teacher literacy recognition based on various algorithmic classification models.

## Comparison of the model training time and test time for evaluating the improvement effect on teacher literacy

The training and testing times serve as metrics to assess the performance of different models within the integrated teacher literacy evaluation model, incorporating classification evaluation algorithms and intelligent devices. The resulting data are illustrated in Figs. 7 and 8, respectively.

Figure 7 illustrates that in the teacher literacy recognition models categorized by different algorithms, the model training time for both algorithms increases in tandem with the number of model iterations. At the 100th iteration, the training and application time for the SVM algorithm model is 5 s, while for the model based on AI and the proposed SOLO method, it is 30 s. As the number of model iterations advances to 600, the training and application times for the SVM-based and SOLO-based models are 128 s and 165 s, respectively. Adhering to the model iteration protocol, the recognition model based on the proposed SOLO algorithm achieves optimal performance after 600 iterations of the algorithm.

In Fig. 8, an increase in the number of model iterations corresponds to an extension of the algorithm's test time. At the 100th iteration, the teacher literacy improvement model based on SVM exhibits a test time of 3 s. In contrast, the proposed SOLO-based teacher literacy improvement model has a test time of 12 s. After 600 iterations of the model, the test times for the SVM-based and SOLO-based teacher literacy improvement models reached 37 s and 52 s, respectively. Consequently, it can be inferred that the proposed

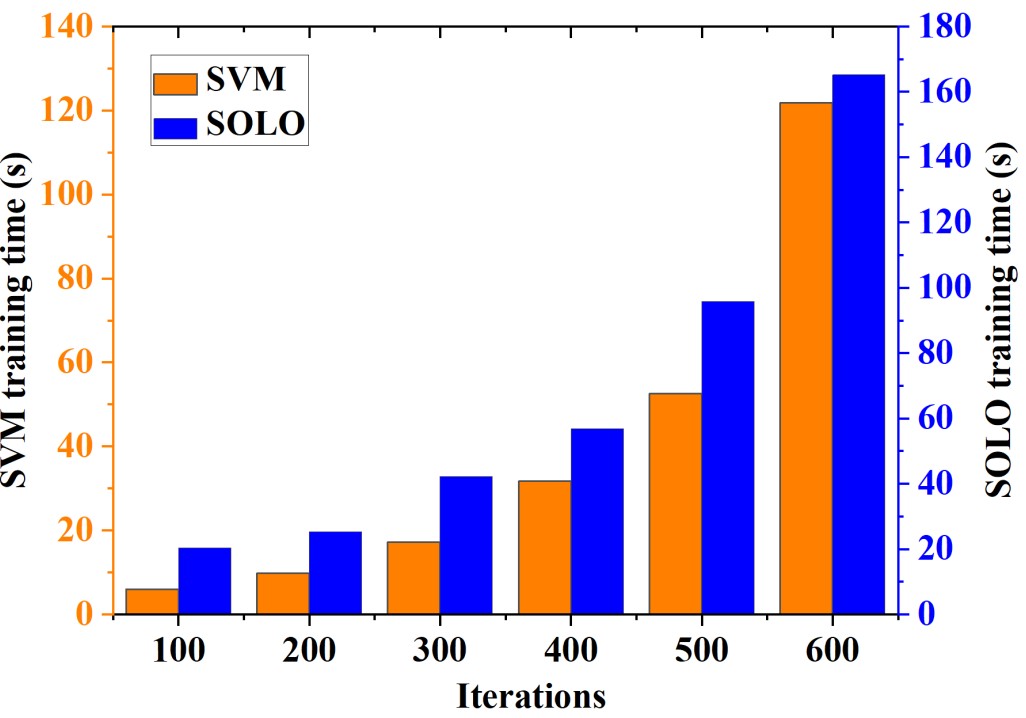

**Figure 7** Training time curve of teacher literacy recognition model classified by different algorithms.

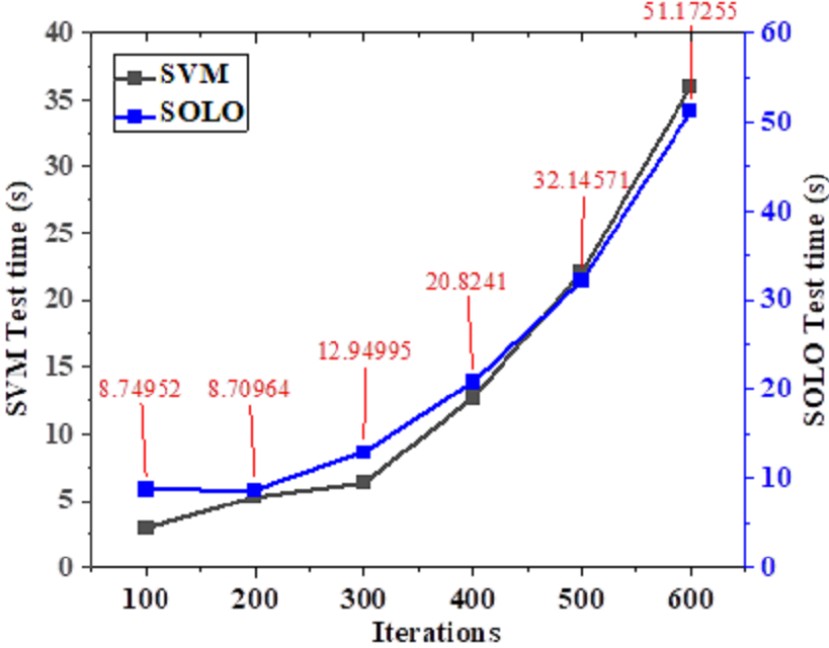

**Figure 8** Test time curve of teacher literacy recognition models classified by different algorithms.

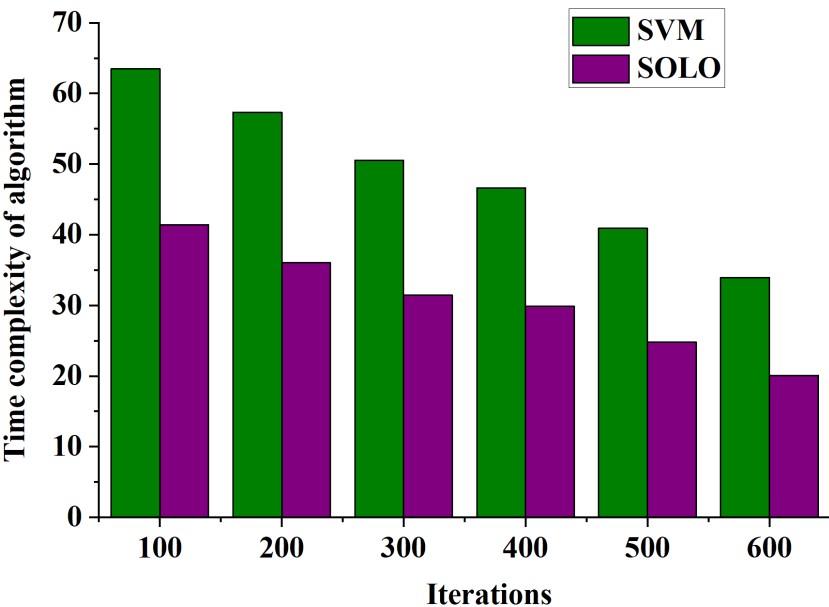

**Figure 9** Change curve of algorithm time complexity of teacher literacy recognition models classified by different algorithms.

SOLO-based algorithm surpasses the SVM model in enhancing the intelligent literacy of middle school teachers.

## Algorithmic complexity analysis of teacher literacy improvement effect

The practical application effect of the SOLO classification evaluation method in enhancing the intelligent literacy of middle school teachers is analyzed. The algorithm's time and space complexity serve as evaluation parameters. The curves depicting the changes in time and spatial complexity for SVM and SOLO algorithms under different model iterations are presented in Figs. 9 and 10, respectively.

Figure 9 illustrates that when comparing the time complexity of SVM-based and SOLO-based algorithms, the time complexity of both algorithms decreases with increased model iterations. At 100 model iterations, the time complexity of the SVM-based and SOLO-based teacher literacy recognition models is 64 and 42, respectively. As the number of iterations reaches 600, the time complexity of the SVM-based recognition model decreases to 38, while the time complexity of the SOLO-based recognition model further reduces to 22. Consequently, the proposed model exhibits superior time complexity performance compared to the SVM-based intelligent literacy improvement model.

Figure 10 reveals that as the number of model iterations increases, the algorithm space complexity of both the SVM-based and SOLO-based intelligent literacy improvement models decreases. At 100 model iterations, the algorithm space complexity of the SVM-based and SOLO-based intelligent literacy improvement models is 63 and 38. When the number of model iterations reaches 600, the spatial complexity of the SVM-based

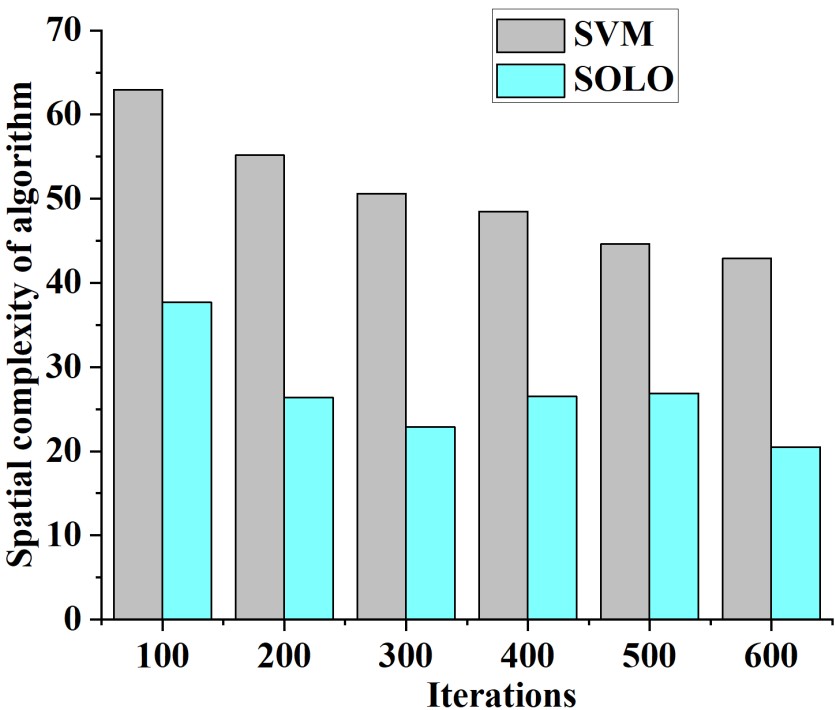

**Figure 10** Change curve of algorithm space complexity of teacher literacy recognition models classified by different algorithms.

and SOLO-based intelligent literacy improvement models is 45 and 22, respectively. Therefore, the comprehensive SOLO-based classification evaluation algorithm contributes to improving the practical application effect of the middle school teachers' literacy recognition model.

## DISCUSSION

By conducting experimental comparisons between the SOLO-based classification evaluation algorithm and the SVM-based intelligent classification model, this study provides a comprehensive analysis of the intelligent literacy of middle school teachers and evaluates the impact on the quality improvement of practical applications. At 600 model iterations, the model based on the SOLO algorithm demonstrates higher accuracy (80%) in teacher literacy recognition compared to the model based on the SVM algorithm (77%). This finding underscores the potential advantages of the SOLO algorithm in enhancing the accuracy of teacher literacy recognition. Under the same number of iterations, the spatial complexity of the SOLO-based intelligent literacy improvement model is lower (22) compared to the SVM-based model (45). This highlights the higher efficiency in resource utilization of the SOLO algorithm while enhancing model performance.

Numerous analogous investigations exist within the scholarly domain. For instance, the work by *Ndukwe & Daniel (2020)* contributed insights into establishing a framework delineating diverse facets of teaching assistants. They formulated a model to deepen

comprehension regarding how teaching assistants contribute to enhancing teaching practices and learning outcomes. Moreover, they advocated for the adoption of a teaching achievement model as a theoretical perspective to guide educators and researchers in leveraging teaching activity data to enhance teaching quality (*Ndukwe & Daniel, 2020*). *Zhao, Wu & Luo (2022)* scrutinized the correlation between dimensions of teachers' AI literacy, aiming to enhance the efficacy of classroom teaching and the integration of AI literacy. Their assessment encompassed teachers' AI literacy levels, covering aspects such as knowledge and understanding of AI, AI application, evaluation of AI application, and AI ethics. Findings indicated that AI applications significantly positively influenced the other three dimensions (*Zhao, Wu & Luo, 2022*). Additionally, *Chen et al. (2022)* employed a structural equation model and fuzzy set for qualitative comparative analysis to investigate the impact of school information and communication technology (ICT) infrastructure and teachers' information literacy on teacher burnout. Their exploration delved into the comprehensive effects of these factors, revealing that constructing high-quality hardware facilities and software resources and implementing advanced school ICT-related policies can mitigate teacher burnout, with teacher information literacy playing a mediating role (*Chen et al., 2022*). Comparatively, the application and evaluation model of teacher intelligent literacy based on the SOLO classification method outperforms the model relying on SVM concerning training time, accuracy, test time, and algorithmic complexity. This observation underscores the superior performance and potential application of the SOLO-based model in the evaluation of teacher intelligent literacy.

In summary, this study possesses both innovative and practical value. For instance, the teacher intelligent literacy assessment model adopts the advanced SOLO classification algorithm, demonstrating higher accuracy and applicability compared to traditional models. Moreover, the experiment's broad applicability at the application level allows for an in-depth analysis of overall teaching discourse, providing a novel evaluation approach for the education domain. Furthermore, a unique solution to address traditional teacher training and evaluation issues is proposed, effectively enhancing the precision and comprehensiveness of teaching quality assessment through the integration of advanced technologies and methodologies.

While the experiment highlights promising aspects of the proposed model, acknowledging and discussing potential limitations is crucial. For example, observed differences in accuracy and processing time may be influenced by specific features of the dataset used. Additionally, the employed AI algorithms may exhibit inherent biases in training data, impacting the model's generalizability. Therefore, as AI continues to evolve in the education domain, future research will focus on several directions to better understand and address ethical challenges in the responsible use of AI in education.

On the one hand, there will be a deeper focus on the ethical considerations of intelligent algorithms in education, encompassing issues such as data privacy, security, algorithmic fairness and transparency, and ethical principles during the learning process. On the other hand, to better meet the needs of the education sector, future research will concentrate on optimizing the SOLO algorithm, including enhancing algorithm performance and designing more adaptive learning models. Regarding the application of AI in education,

future research will focus on the formulation of international regulations and standards, as well as the clear delineation of legal responsibilities, ensuring compliance in global applications. Simultaneously, attention will be given to greater involvement and training of educators in AI, encompassing the design of ethical training programs and studying patterns of educator participation. Research in these directions will provide guidance for the sustainable development of AI applications in the education field, ensuring feasibility and compliance in ethical and legal aspects.

## CONCLUSIONS

The SOLO classification algorithm, coupled with integrated intelligent equipment, finds widespread application across diverse sectors, including education, healthcare, transportation, and more. This study designs two models, namely the SVM-based classification algorithm model and the SOLO-based application model, for evaluating teachers' intelligent literacy. These models undergo performance comparison, utilizing the SOLO classification and evaluation fusion algorithm in conjunction with intelligent teaching equipment for teachers' intelligent literacy training. The results indicate that at a model iteration of 100, the SVM-based teacher literacy recognition achieves an accuracy of 69%. In contrast, the SOLO model exhibits a superior accuracy of 70% in recognizing teacher literacy. Furthermore, when the model iteration reaches 100, the SVM algorithm model necessitates a shorter application and training time of 5s, while the model based on AI and SOLO proposed requires a more extended training and application time of 30s.

Through the application of the SOLO algorithm, teachers' intelligent literacy can be more accurately assessed, providing robust support for the feasibility of introducing advanced technology into the education sector. The model based on the SOLO algorithm exhibits outstanding performance in the experiment. This implies not only that secondary school teachers may benefit from a more efficient and personalized training experience in intelligent literacy training but also that more accurate teaching assessments can be obtained. This holds significant implications for teachers' professional development and the enhancement of teaching quality. The primary contribution and innovation lie in proposing a feasible model for evaluating teachers' intelligent literacy by integrating the advanced SOLO algorithm and intelligent devices. In comparison to the traditional SVM model, the SOLO model demonstrates a significant improvement in accuracy, further validating the research hypothesis. Moreover, the proposed method has broad applicability at the application level, offering a novel assessment approach that delves into the overall discourse of teaching in the education sector. Additionally, although the SOLO model exhibits relatively longer training and application times, its accuracy advantage provides ample compensation for this slight time cost.

Despite achieving a series of encouraging outcomes in the assessment of teachers' intelligent literacy, the study must honestly confront certain limitations. For instance, the data used in the research may be constrained by specific regions or schools, potentially failing to cover all teaching scenarios. Future research should consider expanding data sources and scale to enhance the universality of the study. While the proposed model

performs well under specific conditions, its adaptability needs validation in a more extensive range of teaching environments. Subsequent research should focus on further testing and improving the model to ensure its robustness across various teaching contexts. Moreover, the study is limited by the current state of technology, particularly in speech recognition and behavioral coding. Future research needs to pay attention to the development of emerging technologies to enhance the model's accuracy and efficiency. Simultaneously, future research will place increased emphasis on ethical considerations, particularly addressing the ethical challenges that may arise when AI is applied in the educational process. Considering potential legal restrictions on the use of AI in education in some countries, future research will cautiously consider and adhere to ethical standards to ensure the legal and compliant use of AI in education.

### Funding
This study was supported by the Scientific Research Fund of Tangshan Normal University: Research on the anti-war thoughts of Japanese literary giant Shiga Naoya against the Japanese war of aggression against China (Project No.: 2020B01), Scientific research projects of the "Fourteenth Five-Year Plan" for educational science research in Hebei Province: Research on the improvement of intelligence education literacy of primary and secondary school teachers in the era of artificial intelligence (Project No.: 2203095) and Hebei Higher Education Teaching Reform Research and Practice Project: Research and practice of college English hybrid teaching mode based on smart teaching cloud platform (Project No.: 2020GJJG404). The funders had no role in study design, data collection and analysis, decision to publish, or preparation of the manuscript.

### Grant Disclosures
The following grant information was disclosed by the authors:
Scientific Research Fund of Tangshan Normal University: 2020B01.
Fourteenth Five-Year Plan: 2203095.
Hebei Higher Education Teaching Reform Research and Practice Project: 2020GJJG404.

### Competing Interests
The authors declare there are no competing interests.

### Author Contributions
- Yixi Zhai conceived and designed the experiments, performed the experiments, analyzed the data, performed the computation work, prepared figures and/or tables, and approved the final draft.
- Liqing Chu conceived and designed the experiments, performed the experiments, analyzed the data, performed the computation work, prepared figures and/or tables, and approved the final draft.

- Yanlan Liu conceived and designed the experiments, performed the experiments, analyzed the data, performed the computation work, prepared figures and/or tables, and approved the final draft.
- Dandan Wang analyzed the data, performed the computation work, authored or reviewed drafts of the article, and approved the final draft.
- Yufei Wu performed the computation work, authored or reviewed drafts of the article, and approved the final draft.

## Data Availability

The code is available in the Supplementary Files.

## Supplemental Information

Supplemental information for this article can be found online at http://dx.doi.org/10.7717/peerj-cs.1844#supplemental-information.

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
