# Peer review of "Using deep learning-based artificial intelligence electronic images in improving middle school teachers’ literacy"

_PeerJ Computer Science, doi:10.7717/peerj-cs.1844_

## Round 0.1 · original submission · Major Revisions

Please consider the comments from the reviewers.

**Language Note:** The review process has identified that the English language must be improved. PeerJ can provide language editing services - please contact us at [email protected] for pricing (be sure to provide your manuscript number and title). Alternatively, you should make your own arrangements to improve the language quality and provide details in your response letter. – PeerJ Staff

·

Basic reporting

The article addresses the interesting issue of using the SOLO algorithm to evaluate pedagogical work and improve it. The authors have reviewed a sufficient number of studies on the use of artificial intelligence in various fields, but the analysis is based mostly on regional studies and, accordingly, can be improved by analyzing the world experience, especially the use of artificial intelligence in psychological and pedagogical research. The article has a sufficient number of illustrative materials that give an understanding of the research logic, they are duplicated in separate files, as well as the source data, which meets the requirements. However, more attention should be paid to the use of English to describe the current study by the authors, to improve the level of perception of the material presented.

Experimental design

I thank you for providing the raw data and describing the experimental work, but there are a few things I would like to draw your attention to in order to improve the study:
- It is not very clear from the text of the article what is the basis for training a neural network.
- It also needs to be clarified how exactly the SOLO method is used to train a neural network.
- It is not clear from the text of the article what criteria are used to evaluate the correctness of the lesson, how were they selected?
I would also like to emphasize that the source code is only the beginning: data analysis and deep learning libraries are announced, there is a TeacherIntelligenceModel class, but there is no SOLO code, which makes it impossible to assess the full extent of the research.

Validity of the findings

The validity of the conclusions needs to be brought in line with the text of the article, and it is desirable to reflect the main confirmations of the hypothesis that the authors of the study have so persistently proved. I would also like to suggest adding material on the prospects for further research, especially given the ethical use of artificial intelligence in the educational process, which is not always open information and in some countries is a violation of the law.

Reviewer 2 ·

Basic reporting

The theme and subject presented in this paper are adequate for the journal. I believe the topic is interesting, but in my opinion, this manuscript needs revision, and the detailed comments are listed as below:
- I would recommend to the authors sending the manuscript to an expert in English editing and academic writing for proofreading.
- Although your abbreviations and acronyms will be familiar to many readers, in the spirit of precision and clarity, I would suggest defining them on first usage (see SOLO, ANOVA etc.).
- The authors have presented several problems with traditional teacher training and evaluation; however, it is important to cite the sources that have brought attention to these concerns.
- I recommend that the related work section highlights how artificial intelligence is used in education. I suggest that the authors analyze whether discussing the adoption of AI in healthcare is within the scope of their paper.
- The similarity index is high (29%), I recommend reducing it as much as possible.

Experimental design

Please detail the overall discourse analysis presented in application layer.
The authors should provide a clear discussion characterizing their work, its benefits and drawbacks, how they addressed the problems with their proposed solution. Please state clearly and precisely in the paper what makes this work original.

Validity of the findings

Kindly state what could be the limitations and challenges of your study.
The main novelty and contribution of needs must be summarized and highlight the recommendations based on obtained results. These results are the hallmark for future extension, therefore, please spend some more time writing the conclusion and based on the results suggest new directions.

·

Basic reporting

The paper is sound. The paper provides a clear structure, including Introduction, Methods, Results, Discussion, and Conclusions. The language is generally clear, and the inclusion of references to related studies strengthens the research context. However, as mentioned in the following feedback, minor improvements can be made to enhance clarity, organization, and completeness in specific sections. Once those suggestions are addressed, the paper should be well-rounded in reporting.

The introduction of the research paper provides a comprehensive overview of the background and the problem addressed. However, there are a few suggestions to enhance clarity and coherence:

The introduction touches on the problems with traditional teacher training and evaluation methods, but it could benefit from a more explicit statement of the problem. Clearly articulate middle school teachers' challenges and the current system's limitations.
Improve the transition between the general discussion on the growth of electronic education resources and the specific challenges middle school teachers face. This will make the narrative more cohesive.
Define terms like "intelligent literacy" and briefly explain concepts such as the SOLO algorithm. This will ensure that readers, especially those unfamiliar with the field, can follow the paper more easily.
Some sentences could be rephrased for clarity and conciseness. For example, "The traditional teaching model has yet to meet the diversified learning needs of today's students and the changes in information acquisition (Vadim, 2018)" could be clarified for better readability.

Emphasize the practical implications of the study in the introduction itself. Explain how the proposed use of AI electronic images and the SOLO algorithm addresses the identified problems and improves middle school teachers' training and evaluation.

Experimental design

A few suggestions can enhance the clarity and rigor of the section.

Begin the section with a brief explanation of what AI electronic images entail to provide context for readers who may not be familiar with the term. This will ensure a clear understanding of the technology being employed.
Explain the rationale behind choosing AI electronic images with the SOLO classification method. Briefly explain how AI electronic images support the goals of evaluating teachers' intelligent literacy.
Consider breaking down the section into subsections for each aspect of AI electronic image application (training content, teaching process, and evaluation/feedback). This will enhance the organization and make it easier for readers to navigate.
When discussing the advantages of the SOLO classification method compared to other algorithms like decision trees and SVM, provide a brief rationale or context for why these particular algorithms were chosen for comparison. This will help readers understand the significance of the comparison.
Acknowledge and discuss any potential limitations of the proposed model, such as constraints in the data types that can be effectively processed or potential biases in the AI algorithms. This adds transparency to the research.
When discussing the experimental validation, explicitly define the metrics used for evaluation, such as accuracy, time complexity, and space complexity. Ensure that readers understand the significance of these metrics in assessing the model's performance.
Conclude the experimental validation section by summarizing the essential findings and implications. Discuss how the results contribute to understanding the proposed model's effectiveness.

Validity of the findings

Ok

Additional comments

In the conclusions, while summarizing the findings, consider providing more context on the practical implications of the results. Discuss how the superior performance of the SOLO-based model translates into potential benefits for middle school teachers, such as more efficient training and accurate evaluation. Additionally, emphasize the relevance of these findings for the broader field of education and the potential impact on teaching practices.
Address the identified limitations more explicitly. Acknowledge the challenges associated with subjective and objective conditions in teaching classification and effect evaluation, and discuss potential strategies or future research directions to mitigate these limitations. This will provide a more transparent and nuanced interpretation of the study's outcomes.

---

## Round 0.2 · accepted · Accept

The authors considered all reviews, and it can be accepted.

Reviewer 2 ·

Basic reporting

The changes are satisfactory.

Experimental design

I have reviewed this article before. It has improved as compared to the previous state.

Validity of the findings

The authors have addressed all my concern in this revised version, and made afford to improve the quality of the manuscript.